# Learn What You Need in Personalized Federated Learning

## Abstract

Personalized federated learning aims to address data heterogeneity across local clients in federated learning. However, current methods blindly incorporate either full model parameters or predefined partial parameters in personalized federated learning. They fail to customize the collaboration manner according to each local client's data characteristics, causing unpleasant aggregation results. To address this essential issue, we propose *Learn2pFed*, a novel algorithm-unrolling-based personalized federated learning framework, enabling each client to adaptively select which part of its local model parameters should participate in collaborative training. The key novelty of the proposed *Learn2pFed* is to optimize each local model parameter's degree of participant in collaboration as learnable parameters via algorithm unrolling methods. This approach brings two benefits: 1) mathmatically determining the participation degree of local model parameters in the federated collaboration, and 2) obtaining more stable and improved solutions. Extensive experiments on various tasks, including regression, forecasting, and image classification, demonstrate that *Learn2pFed* significantly outperforms previous personalized federated learning methods.

## 1 Introduction

Federated learning (FL) is an emerging collaboration paradigm that was first introduced in (McMahan et al., 2017). Since only an update to the current global model is uploaded in FL, instead of raw datasets, it can protect data privacy. Due to this characteristic, it is widely used in finance (Long et al., 2020), healthcare (Nguyen et al., 2022), smart cities (Zheng et al., 2022), and other fields. However, data heterogeneity across local clients creates deviations between local models and the global model so that they cannot reach the consensus (Wang et al., 2020). Hence, personalized federated learning (Tan et al., 2022; Kulkarni et al., 2020) has been explored to train improved local models within the federated learning framework, instead of relying solely on a global model.

Previous research in personalized Federated Learning (FL) has traditionally emphasized training local models with all parameters, as shown in Figure 1(a). This has been achieved through two predominant approaches. Finetuning-based personalized FL methods adapt the local model from the global model, in line with established FL techniques such as FedAvg (McMahan et al., 2017) and FedProx (Li et al., 2020a), utilizing finetuning (FT) across all local model parameters. Similarly, model-based personalized FL methods (Li et al., 2021; T Dinh et al., 2020) have garnered popularity by introducing regularization terms with adjustable hyper-parameters to balance the relationship between local models and the global model. In addition, recent studies have found that involving only a subset of parameters in collaboration can yield better results than involving full parameters in FL, relying on heuristic parameter selection (Arivazhagan et al., 2019; Collins et al., 2021; Pillutla et al., 2022) or binary decision learning (Setayesh et al., 2022; Isik et al., 2023), as illustrated in Figure 1(b). However, these works do not examine to what degree these chosen partial parameters should be integrated into the federated learning process. The limited variability caused by binary selection hinders the creation of personalized models that could better adapt to local data.

Motivated by this, we aim to learn to determine which part of a local model should participate in federated learning and further to *what degree*, as illustrated in Figure 1(c). To achieve this, our key idea is to consider each parameter's degree of participant in collaboration as one learnable variable, and then optimizes those parameters in algorithm unrolling. Following this spirit, we propose a novel

algorithm-unrolling-based personalized federated learning framework, *Learn2pFed*. Specifically, it unrolls the parameters, originally in the iterative algorithm that can indicate the degree of participant in collaboration, into layers of a deep network. Supervised by the sum of training losses collected from all local clients, *Learn2pFed* adaptively learns the characteristics of the local data and select appropriate partial parameters.

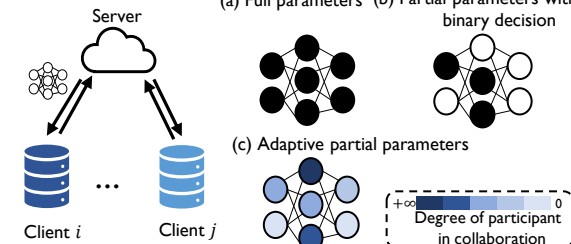

Compared to previous works, *Learn2pFed* has two distinct advantages: 1) it dynamically determines which parameters of the local model need to collaborate in FL and to what degree, thus adapting to the local data better and improving the performance of personalized FL; and 2) it leverages algorithm unrolling to make hyper-parameters learnable and significantly improves the model capability.

Figure 1: Three federated ways of local model parameters: sending (a) full parameters; (b) partial parameters with binary decision; (c) adaptive partial parameters. We aim to determine the part and the degree of local model parameters that participate in federated collaboration.

To evaluate *Learn2pFed*, we consider various personalized FL tasks including regression, forecasting and image classification on different datasets: synthetic polynomial data, power consumption data, Fashion-MNIST (Xiao et al., 2017) and CIFAR-10 (Krizhevsky et al., 2009). *Learn2pFed* outperforms the previous personalized FL methods in the above three tasks.

Our main contributions are three-fold: (1) We introduce adaptive collaboration in personalized federated learning by enabling each client to select which part of its local model parameters should participate in personalized federated learning, addressing data heterogeneity and improving aggregation results. (2) We propose a novel algorithm-unrolling-based framework *Learn2pFed* for personalized federated learning to optimize the degree of participant for each model parameter in collaboration, which turns the fixed hyper-parameters in the optimization into learnable parameters in our framework. (3) We conduct extensive experiments in various tasks, and show that the performance is competitive with state-of-the-art methods.

## 2 RELATED WORKS

**Vanilla federated learning:** Federated learning (McMahan et al., 2017) enables multiple local clients to collaboratively train a global model without sharing their raw data. The basic idea is to distribute the training process across clients, with each training the model on its local data. The local models are then aggregated into a global model in the server, which is then sent back to the local clients for further training. This process is repeated iteratively until the global model converges to a satisfying level of accuracy. It assumes that the datasets from different clients are sampled from the same distribution, however, such data homogeneity assumption does not hold in practice.

**Personalized federated learning:** Personalized federated learning (Tan et al., 2022) aims to address data heterogeneity across local clients in federated learning by mainly two approaches: personalizing the global model or learning the personalized model. The first approach focuses on generalizing the global model and local adaptation. It involves training a single global model that is then applied in downstream tasks using techniques like finetuning and knowledge transfer (Kairouz et al., 2021). While effective, this approach may not fully capture the unique characteristics of individual clients' data. The second approach, which our method belongs to, aims to provide personalized solutions within the federated learning framework. By modifying the learning process with full model parameters in FL, these personalized FL methods are presented in a variety of ways through optimization on the well-designed objective functions (T Dinh et al., 2020; Li et al., 2021; Lin et al., 2022), meta-learning (Fallah et al., 2020), clustering (Sattler et al., 2020; Marfoq et al., 2022), generative networks (Shamsian et al., 2021), etc. In addition, some works (Arivazhagan et al., 2019; Collins et al., 2021; Pillutla et al., 2022) realize that personalization with full parameters may be unnecessary, and manually divide them into personal parameters and shared parameters, where only the former is updated locally. Another branch of works enables clients to selectively share a dynamic parameter subset, including methods such as parameter pruning (e.g., FedClip (Lu et al., 2023),

FedMP (Jiang et al., 2022)), and subnet training (e.g., HeteroFL (Diao et al., 2021), FedPM (Isik et al., 2023)). Rather than learning pruning parameters to determine the shared part of model in a binary manner, our proposed method leverages algorithm unrolling to transform the traditionally fixed parameters into learnable ones, enabling continuous refinement of model personalization.

**Algorithm unrolling:** Algorithm unrolling (Monga et al., 2021) is a technique that unrolls one specific iterative optimization algorithm, e.g., the iterative shrinkage and thresholding algorithm (ISTA (Beck & Teboulle, 2009) ), the alternating direction method of multipliers (ADMM (Boyd et al., 2011)), into stacked layers of a deep network. Then, each forward propagation of the network is equivalent to performing several iterations of the iterative algorithm. In this way, unrolling enhances both the representation ability of the iterative algorithm and the generalization ability of the generic neural networks, thus reaching an attractive balance. For these advantages, it has been widely applied in various domains, including the context of sparse coding (Gregor & LeCun, 2010), compress sensing (Yang et al., 2018) and signal denoising (Chen et al., 2021; Vu et al., 2021; Li et al., 2020b). In our work, we leverage deep unrolling to determine the personal parameters in personalized federated learning, bridging the gap between iterative algorithms and the federated learning framework.

## 3 PRELIMINARY

The personalized federated learning framework consists of one parameter server and $M$ local clients, where the $i$-th client holds the local data $\mathcal{D}_i = \{X_i, Y_i\}$ with $X_i \in \mathbf{R}^{n_i \times k}, Y_i \in \mathbf{R}^{n_i}$ generated from one of the unknown models. $n_i$ denotes the number of samples in the $i$-th client and $k$ denotes the feature dimension. Let $w \in \mathbf{R}^k$ be the global model parameters, and $v_i \in \mathbf{R}^k$ be the $i$-th local model parameters for $i \in [M]$, where we denote the set $\{1, 2, \ldots, M\}$ for any integer $M$ as $[M]$.

Generally, the objective of personalized FL is formed as the local objectives $L_i(v_i; w^*)$ given the optimized global model $w^*$, composed of local empirical loss $F_i(v_i)$ on the local training data in the $i$-th client and the regularized term $\|v_i - w^*\|^2$ indicating the distance between the global model and local model. Mathematically, the optimization of personalized FL is typically formed as below.

$$\min_{v_i} \quad L_i(v_i; w^*) = F_i(v_i) + \lambda \|v_i - w^*\|^2, \quad \text{s.t. } w^* = \arg\min_w \sum_{i=1}^{M} p_i L_i(v_i^*; w), \quad (1)$$

where $\lambda$ and $\{p_i\}$ are two kinds of positive hyper-parameters in personalized FL, and $\{\cdot\}$ denotes the abbreviation of $\{\cdot\}_{i=1}^M$. Specifically, $\lambda$ regularizes the similarity between the global model and local models, with larger values of $\lambda$ indicating stronger similarity. When $\lambda \to \infty$, personalized FL degrades to the general FL; when $\lambda = 0$, personalized FL degrades to the local independent learning.

While (1) provides the mathematical form commonly used in personalized federated learning methods, it has a limitation arisen from treating the entire local parameter model as a single entity, thus overlooking the unique characteristics of local data. This limitation may hamper the ability to adapt the model to individual data distributions and can result in worse performance in personalized federated learning. Therefore, it becomes crucial to address this limitation and develop a solution by learning the specific parameters of local models in collaboration.

In this regard, we propose *Learn2pFed*, a novel framework that entails redesigning the formulation of (1). We will delve into the details of the *Learn2pFed* framework in the next section.

## 4 *Learn2pFed*: UNROLLING-BASED PERSONALIZED FL FRAMEWORK

To determine which specific parameters of the local models should participate in the federated learning, this section introduces *Learn2pFed*, a novel deep unrolling framework for personalized federated learning from both aspects of mathematical optimization and federated implementation. We further discuss its characteristics including parameters, communication and privacy.

### 4.1 OVERALL OPTIMIZATION

Based on the original optimization problem (1), we introduce another crucial component, $\Lambda_i$, alongside the aggregation weight variable $p_i$. This addition allows us to achieve personalized regularization for each model parameter, further enhancing the adaptive federated aggregation.

**Regularized variable** $\Lambda_i$**:** Instead of using a scalar $\lambda$ in (1) to regularize all model parameters, we introduce a personalized diagonal matrix $\Lambda_i \in \mathbf{R}^{k \times k}$ for the $i$-th client for element-wise regularization. Each element $(\Lambda_i)_{jj}$ is a positive value, indicating the degree of each model parameter that participates in the federated collaboration. This matrix enables customized regularization for different parameters within each client's local model. Such fine-grained personalized regularization allows for adaptive control of the degree of participant in collaboration, improving model performance by tailoring the regularization to the specific characteristics of each client's data.

Subsequently, the overall optimization of *Learn2pFed* is formulated as a bi-level optimization problem, which involves the learning objective $\mathcal{P}_f(\{v_i\}, w)$ and the constraint problem $\mathcal{P}_b(\{\Lambda_i, p_i\})$:

$$\min_{\{v_i\}, w} \quad \mathcal{P}_f(\{v_i\}, w) = \frac{1}{M} \sum\nolimits_{i=1}^{M} p_i \left( F_i(v_i) + (v_i - w)^\top \Lambda_i (v_i - w) \right)$$

$$\text{s.t.} \quad \{\Lambda_i, p_i\} = \arg \min_{\{\Lambda_i, p_i\}} \mathcal{P}_b(\{\Lambda_i, p_i\}) = \sum\nolimits_{i=1}^{M} F_i(v_i^\star), \quad (2)$$

where $v_i^\star$ is the output of $\mathcal{P}_f(\{v_i\}, w)$ and $F_i(v_i^\star)$ denotes the local training loss in the $i$-th client based on the specific tasks, such as Mean-Squared-Error (MSE) loss for regression or Cross-Entropy (CE) loss for classification. Intuitively, (2) aims to output the learned local model $\{v_i\}$ for personalized FL, while learning the adaptive collaboration pattern via learnable parameters $\{\Lambda_i, p_i\}$ with the supervised information in the form of the sum of local training losses. Unlike (1), (2) also includes the learning of $\{\Lambda_i, p_i\}$, thus it can adaptively determine the specific part of local model parameters involved in the collaboration, allowing for a more flexible and effective personalized federated learning process. To address the optimization problem presented in (2), we leverage algorithm unrolling. Specifically, our approach involves solving the objective of (2) using a single optimization algorithm, as discussed in Section 4.2. Subsequently, we unroll this algorithm into layers and train a deep network, as explained in Section 4.3.

## 4.2 Optimization Algorithm

This sub-section aims to solve the learning objective of (2) with fixed parameters $\{\Lambda_i, p_i\}$. Since the global model and local models are coupled in $\mathcal{P}_f(\{v_i\}, w)$ in (2), the alternating direction method of multipliers (ADMM (Boyd et al., 2011)) is a way to split the variables into local sides and the global side. Specifically, we introduce the auxiliary variable $\{z_i\}$ indicating the consensus constraint in the local. It brings two benefits: 1) it decouples the global and local model so that solving the local variables can be carried out in parallel in each client; 2) it allows for a more flexible expression of constraints making the problem easier to solve. Then, $\mathcal{P}_f(\{v_i\}, w)$ in (2) is reformulated as below.

$$\min_{\{z_i\}, \{v_i\}, w} \quad \mathcal{P}_f(z_i, v_i, w) = \frac{1}{M} \sum\nolimits_{i=1}^{M} p_i \left( F_i(v_i) + z_i^\top \Lambda_i z_i \right) \quad \text{s.t.} \ z_i = v_i - w. \quad (3)$$

For faster convergence, we also provide its augmented Lagrangian as

$$\mathcal{L}_{p_i, \rho_i, \Lambda_i}(\{v_i\}, \{z_i\}, w; \{\alpha_i\}) = \frac{1}{M} \sum\nolimits_{i=1}^{M} p_i \left( F_i(v_i) + z_i^\top \Lambda_i z_i + \frac{\rho_i}{2} \|z_i - v_i + w + \alpha_i\|^2 \right), \quad (4)$$

where $\{\alpha_i\}$ are Lagrangian multipliers in the local and $\{\rho_i\}$ are positive hyper-parameters. That is, taking the regression problem as example where $F_i(v_i) = \|X_i v_i - Y_i\|^2$, the ADMM alternatively optimizes $\{v_i\}, \{z_i\}, w, \{\alpha_i\}$ by solving the following sub-problems in the $\ell$-th iteration.

$$v_i^\ell := \arg \min_{v_i} \quad \|X_i v_i - Y_i\|^2 + \frac{\rho_i}{2} \left\| z_i^{\ell-1} + w^{\ell-1} + \alpha_i^{\ell-1} - v_i \right\|^2, \quad (5)$$

$$z_i^\ell := \arg \min_{z_i} \quad z_i^\top \Lambda_i z_i + \frac{\rho_i}{2} \left\| z_i - v_i^\ell + w^{\ell-1} + \alpha_i^{\ell-1} \right\|^2, \quad (6)$$

$$w^\ell := \arg \min_{w} \quad \sum_i^M \frac{p_i \rho_i}{2} \left\| z_i^\ell - v_i^\ell + w + \alpha_i^{\ell-1} \right\|^2, \quad (7)$$

$$\alpha_i^\ell := \quad \alpha_i^{\ell-1} + \rho_i \left( z_i^\ell - v_i^\ell + w^\ell \right). \quad (8)$$

Since it follows the standard ADMM, its convergence is guaranteed by (Hong et al., 2016; Hong & Luo, 2017). After performing multiple iterations, e.g., $L$ iterations, till convergence as described above, we obtain the local model $\{v_i^L\}$.

However, determining $\{\Lambda_i, p_i, \rho_i\}$ plays a critical role in (5)-(8) since each has a distinct impact on the performance and convergence behavior. For example, the elements of $\Lambda_i$ control the similarity between local and global models for specific features. Tuning these elements influences the models'

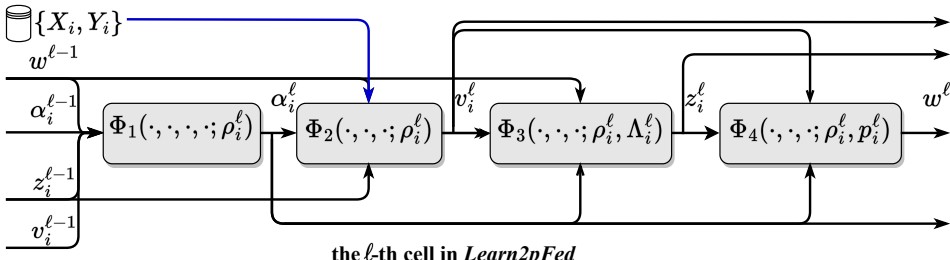

the $\ell$-th cell in *Learn2pFed*

Figure 2: Illustration of the $\ell$-th cell in *Learn2pFed*. It unrolls (9)-(12) into one four-layer cell of the deep network. Black lines indicate the flow of intermediate variables, e.g., $\{w^\ell, \alpha^\ell, z_i^\ell, v_i^\ell\}$ in the $\ell$-th cell. Blue line indicates the local data flow, which however, will not be shared across clients.

behavior in capturing global patterns. Similarly, $\rho_i$ affects the convexity and underfitting of local models. Balancing $\rho_i$ is crucial to avoid overfitting or excessive similarity. But selecting suitable $\{\Lambda_i, p_i, \rho_i\}$ is challenging due to their interplay and sensitivity. Manual tuning is time-consuming and prone to biases. Moreover, directly optimizing them in the original problem is not feasible for the trivial solution. Therefore, how to determine $\{\Lambda_i, p_i, \rho_i\}$ is a big challenge, and we provide our method making them learnable in the next section.

### 4.3 ALGORITHM UNROLLING

We introduce the proposed personalized FL framework *Learn2pFed* based on algorithm unrolling to adaptively determine the learnable parameters in the above section. The key idea is to view the parameters $\Theta^\ell = \{\Lambda_i^\ell, p_i^\ell, \rho_i^\ell\}$ in (5)-(8) as trainable parameters in a deep network with the input and parameters as $\Phi(\{X_i, Y_i\}; \{\Theta^\ell\}_{\ell=1}^L)$, where local data $\{X_i, Y_i\}$ are privately stored in local clients. Specifically, we solve the optimization in (5)-(8) iteratively, and model one of iterations as a four-layer cell in $\Phi(\{X_i, Y_i\}; \{\Theta^\ell\}_{\ell=1}^L)$, as illustrated in Figure 2. Mathematically, we provide the formulations of the intermediate outputs in the $\ell$-th cell as follows.

$$\alpha_i^\ell \leftarrow \Phi_1(\alpha_i^{\ell-1}, v_i^{\ell-1}, z_i^{\ell-1}, w^{\ell-1}; \rho_i^\ell) = \alpha_i^{\ell-1} + \rho_i^\ell(z_i^{\ell-1} - v_i^{\ell-1} + w^{\ell-1}), \tag{9}$$

$$v_i^\ell \leftarrow \Phi_2(\alpha_i^\ell, z_i^{\ell-1}, w^{\ell-1}; \rho_i^\ell) = (X_i^\top X_i + \rho_i^\ell \mathcal{I}_k)^{-1}(\rho_i^\ell(w^{\ell-1} + z_i^\ell + \alpha_i^\ell) + X_i^\top Y_i), \tag{10}$$

$$z_i^\ell \leftarrow \Phi_3(\alpha_i^\ell, v_i^\ell, w^{\ell-1}; \rho_i^\ell, \Lambda_i^\ell) = \rho_i^\ell(\text{ReLU}(\Lambda_i^\ell) + \rho_i^\ell \mathcal{I}_k)^{-1}\left(v_i^\ell - w^{\ell-1} - \alpha_i^\ell\right), \tag{11}$$

$$w^\ell \leftarrow \Phi_4(\alpha_i^\ell, v_i^\ell, z_i^\ell; \rho_i^\ell, p_i^\ell) = \frac{\sum_i p_i^\ell \rho_i^\ell \left(v_i^\ell - z_i^\ell - \alpha_i^\ell\right)}{\sum_i p_i^\ell \rho_i^\ell}, \tag{12}$$

where $\mathcal{I}_k$ means the identity matrix with the dimension $k$, and the parameters $\Theta^\ell = \{\Lambda_i^\ell, p_i^\ell, \rho_i^\ell\}$ are learnable. This is the main difference from the iterative algorithm in (5)-(8). In addition, we build up the ReLU (LeCun et al., 2015) module to guarantee the diagonal element of $\{\Lambda_i^\ell\}$ is positive, which is given manually in the previous. In this way, the proposed *Learn2pFed* concatenates multiple four-layer modules as described above into the deep network. It is worth noting that the update sequence of the ADMM has little impact on its convergence, hence the decision to design the layers is for easier federated implementation.

In the training stage of *Learn2pFed*, we consider the following optimization problem:

$$\min_{\{\Lambda_i, p_i, \rho_i\}} \mathcal{P}_b(\{\Lambda_i, p_i, \rho_i\}) = \sum_{i=1}^M F_i(v_i^L), \tag{13}$$

where $F_i(v_i^L)$ is the local training loss based on the output of the final layer. In contrast to $\mathcal{P}_b(\{\Lambda_i, p_i\})$ in (2), $\{\rho_i\}$ introduced by the ADMM is also treated as the target variable in (13). Then, the parameters $\{\Lambda_i, p_i, \rho_i\}$ are updated iteratively through the standard gradient descent.

In conclusion, the proposed *Learn2pFed* framework performs the iterative algorithm in forward propagation, and trains the learnable parameters in the deep network supervised by the sum of local training losses, which carries high-level information from other clients. *Learn2pFed* enjoys the following benefits: 1) it adaptively learns $\{\Lambda_i, p_i, \rho_i\}$ during the training process, enabling it to determine the degree of participation of each local model's parameters in the collaboration. This adaptive learning capability allows the framework to dynamically adjust the collaboration strategy based on the specific characteristics of the data and the optimization problem at hand. 2) the

integration of deep neural networks in *Learn2pFed* provides a powerful modeling capability. By leveraging the expressive power of deep networks, the framework can capture complex patterns in each local data, leading to improved performance over traditional iterative algorithms and heuristic neural network approaches.

## 4.4 FEDERATED IMPLEMENTATION

We provide a detailed federated implementation of *Learn2pFed*. We initialize local learnable parameters $\{\Lambda_i^{\ell-1}, \rho_i^{\ell-1}\}$, the local model, and its intermediate variables $\{v_i^{\ell-1}, z_i^{\ell-1}, \alpha_i^{\ell-1}\}$ on the client sides, where $\ell = 1$. Additionally, we initialize global learnable parameters $\{p_i^{\ell-1}, \gamma_i^{\ell-1}\}$ and the global model $w^{\ell-1}$, with $\gamma_i^{\ell-1}$ serving as a copy of $\{\rho_i^{\ell-1}\}$ on the server side. We then introduce the implementation on both client and server sides.

1) ***Client-Side Computation and Communication:*** In the client sides, *Learn2pFed* updates the intermediate variables $\{\alpha_i^\ell, v_i^\ell, z_i^\ell\}$ by (9), (10), (11), respectively, in the $\ell$-th cell of the deep network based on the learnable parameters $\{\Lambda_i^\ell, \rho_i^\ell\}$. Note that when updating $\{v_i^\ell\}$, since $F_i(v_i)$ can be convex or non-convex, we need to discuss the solution separately, and take the two tasks that we will face in the experiments for example. In regression tasks, we perform (10) directly. However, in classification tasks, $F_i(v_i)$ is non-linear. Then we reformulate the update of $\{v_i^\ell\}$ in (10) using the gradient descent as follows.

$$v_i^\ell \leftarrow v_i^{\ell-1} - lr * \partial h_i(v_i)/\partial v_i, \tag{14}$$

where we denote $h_i(v_i) = F_i(v_i) + \frac{\rho_i^\ell}{2} \left\| z_i^{\ell-1} + w^{\ell-1} + \alpha_i^\ell - v_i \right\|^2$ based on (5). We find that the approximation accuracy of the solution in this layer does not affect the convergence of the network much in practice, so the learning rate $lr$ can be artificially set.

As for the communication, each local client sends the vector $v_i^\ell - z_i^\ell - \alpha_i^\ell$ and the local training loss $F_i(v_i^L)$ to the server in each cell $\ell \in [L]$ and the final cell $L$ of the network, respectively. Additionally, each client receives the global model $w^\ell$ and the sum of local training losses across clients broadcasted by the serveri n each cell $\ell \in [L]$ and the final cell $L$ of the network, respectively. Finally, each client leverages the sum of losses to independently update their learnable parameters $\{\Lambda_i^\ell, \rho_i^\ell\}$ using the gradient descent method in the final cell $L$ of the network in the client sides.

2) ***Server-Side Computation and Communication:*** In the server side, *Learn2pFed* updates the intermediate variable $w^\ell$ by (8) in the $\ell$-th cell of the deep network based on the learnable parameters $\{p_i^\ell, \rho_i^\ell\}$. Since $\{\rho_i^\ell\}$ appears in both sides, we copy it as $\gamma_i^\ell$ and update the only in the server side. In terms of communication, the server broadcasts the updated global model $w^\ell$ and the sum of local training losses back to all the clients in each cell $\ell \in [L]$ ann the final cell $L$ of the network. At the same time, the learnable parameters $\{p_i^\ell, \gamma_i^\ell\}$ are updated based on the sum of local training losses in the server side using the gradient descent method in the final cell $L$ of the network.

Then, the above computations and communications are repeated untill *Learn2pFed* converges. To sum up, we summarize the overall algorithm in Alg. 1 in the Appendix A.2.

**Discussion:** Though FL framework avoids local data being exposed, the full model parameters may still leak the data privacy by various attack methods (Fredrikson et al., 2015; Shokri et al., 2017). In our framework, we send the linear combination of multiple local variables, which protects the local client from such attacks. In addition, the server only collects local losses in the final layer, which also will not leak sensitive information about local data. More discussions go to Appendix A.2.

## 5 EXPERIMENTS

In this section, we first conduct algorithm comparisons in a three-order polynomial regression task, and investigate the characteristics of *Learn2pFed* through ablation studies. Further, we apply it in both power consumption forecasting and image classification with the real-world data in various personalized FL settings, demonstrating superior performance compared to baseline methods.

## 5.1 EXPERIMENTAL SETUP

**Baselines.** We compare our proposed *Learn2pFed* with 12 representative baselines under multiple experimental settings. Local-Only indicates that each client trains an independent model using its local

Table 1: Regression performance w.r.t. three personalized FL settings on synthetic data. The proposed *Learn2pFed* achieves the best performance in all the three personalized FL settings.

| Methods | Type in FL | Averaged RMSE | | |
|---|---|---|---|---|
| | | Setting 1 | Setting 2 | Setting 3 |
| Local-Only | - | 0.0204 | 0.0149 | 0.0208 |
| FedAvg | Generalized | $0.2067 \pm 0.0070$ | $1.6571 \pm 0.1238$ | $3.9092 \pm 3.8794$ |
| FedProx | Generalized | $0.1351 \pm 0.0418$ | $0.4214 \pm 0.4953$ | $2.3072 \pm 1.5184$ |
| FedAvg + FT | Finetune | $0.0023 \pm 0.0001$ | $0.0014 \pm 0.0001$ | $0.0716 \pm 0.0973$ |
| FedProx+ FT | Finetune | $0.0132 \pm 0.0179$ | $0.0176 \pm 0.0245$ | $0.0109 \pm 0.0150$ |
| FedPer | Split layers | $0.0006 \pm 0.0008$ | $0.0016 \pm 0.0007$ | $0.0029 \pm 0.0026$ |
| FedRep | Split layers | $0.0175 \pm 0.0045$ | $0.0136 \pm 0.0023$ | $0.0154 \pm 0.0027$ |
| pFedMe | Optimization | $0.0017 \pm 0.0002$ | $0.0111 \pm 0.0004$ | $0.0113 \pm 0.0008$ |
| Ditto | Optimization | $0.0005 \pm 0.0000$ | $0.0011 \pm 0.0007$ | $0.0004 \pm 0.0000$ |
| lp_proj | Optimization | $0.0023 \pm 0.0000$ | $0.0015 \pm 0.0000$ | $0.0017 \pm 0.0000$ |
| ***Learn2pFed*** | Optimization | $\mathbf{0.0002 \pm 0.0002}$ | $\mathbf{0.0003 \pm 0.0002}$ | $\mathbf{0.0003 \pm 0.0002}$ |

data without federated collaboration. FedAvg (McMahan et al., 2017) and FedProx (Li et al., 2020a) are two general FL baselines, while FedAvg+FT and FedProx+FT are their fine-tuning versions. Other personalized FL baselines include FedPer (Arivazhagan et al., 2019), FedRep (Collins et al., 2021), Ditto (Li et al., 2021), pFedMe (T Dinh et al., 2020), lp_proj (Lin et al., 2022), CFL (Sattler et al., 2020), and KNN-per (Marfoq et al., 2022). Note that cluster-based personalized FL methods like CFL and KNN-per are only used in our classification tasks.

**Training Details.** We consider 500 communication rounds of FL and 2 epochs for each round with the batch size of 64. We use Adam as the optimizer with a learning rate of 0.01. For regression and forecasting tasks, we build up *Learn2pFed* following Alg. 1 with $L = 10$, while using MLP and LSTM (Yu et al., 2019) as baseline models, respectively, for comparison. For image classification tasks, we use *Learn2pFed* as a plug-and-play model that replaces the last layer of the original CNN with a linear approximation; see more details in Appendix B.

## 5.2 POLYNOMIAL REGRESSION TASK

**Dataset and Federated Settings.** In this experiment, each client $i$ has a distinct ground-truth (gt) objective function $f_i(x) = \sum_{d=0}^{3} \boldsymbol{a}_i[d] \cdot x^d$, where $\boldsymbol{a}_i = [a_0, a_1, a_2, a_3]$ is the polynomial coefficient vector. Different clients have different coefficient vectors, while they may share some coefficients. Here, we consider three different settings: Setting 1: all clients share three coefficients, i.e., $\boldsymbol{a}_i[d] = \boldsymbol{a}_j[d]; \forall i, j; \forall d \in \{0, 1, 2\}$. Setting 2: all clients share two coefficients, i.e., $\boldsymbol{a}_i[d] = \boldsymbol{a}_j[d]; \forall i, j; \forall d \in \{0, 1\}$. Setting 3: all clients share one coefficient, i.e., $\boldsymbol{a}_i[0] = \boldsymbol{a}_j[0]; \forall i, j$. The remaining coefficients are set distinctly across clients. Note that since high-order coefficients can have a greater impact on the disturbance of the function, we prefer to keep the lower-order coefficients the same across clients to increase the task's difficulty. Finally, we generate local data by adding Gaussian noise to the local gt function with a mean of 0 and a standard deviation of 0.1.

**Results and Analysis.** We perform the experiments for five independent trials with full 10-client participation, and report the averaged Root-Mean-Squared-Error (RMSE) results in Table 1. We see that (1) the optimization-based methods, including Ditto (Li et al., 2021) and lp_proj (Lin et al., 2022), perform better in terms of both accuracy and stability, as they exhibit smaller RMSEs and variances compared to other approaches. However, the performance of the methods varies depending on the complexity of the dataset. For example, in simpler Setting 1, Ditto and FedPer show better performance than other methods, while in the more complex Setting 3, only Ditto outperforms other methods. This suggests that the choice of the method depends on the characteristics of the dataset. (2) Notably, *Learn2pFed* consistently outperforms other methods in fitting the polynomial model across all experimental settings, indicating that *Learn2pFed* is effective in capturing the underlying patterns in the personalized data and is robust to variations in the input. More visualization results are shown in Appendix D.1.

**Impact of Learnable Parameters.** Table 4 in Appendix C.1 reveals that learning more learnable parameters increases the representation power of *Learn2pFed* and improves the performance. Further,

learning $\{\rho_i\}$ in (8), which play a role like learning rates in forward propagation (9), is shown to be not helpful enough. However, learning the parameters $\{p_i, \Lambda_i\}$ (especially $\{\Lambda_i\}$) plays an important role in *Learn2pFed* since they are more concerned with the FL process.

**Impact of $\{\Lambda_i\}$.** We evaluate the impacts of learning $\{\Lambda_i\}$ by comparing the the performance of *Learn2pFed* achieved with and without learning $\{\Lambda_i\}$ in Setting 1. We find that learning $\{\Lambda_i\}$ greatly improves performance. Specifically, the averaged RMSE is decreased from 0.0026 to 0.0002, a $92\%$ reduction. By further analyzing the learned $\{\Lambda_i\}$ in Figure 3, we see that the element of the matrix $(\Lambda_i)_{33} \to 0$ for all clients, which are consistent with our expectation since $(\Lambda_i)_{33}$ varies across clients in Setting 1 and thus should be learned locally. Besides, it also suggests that $a_i[2]$ (in the ground-truth objective function) need to be learned locally since $(\Lambda_i)_{22} \to 0$. Overall, these findings highlight the importance of learning $\{\Lambda_i\}$ in our *Learn2pFed* algorithm and demonstrate its ability to adapt to the characteristics of the underlying local data distribution.

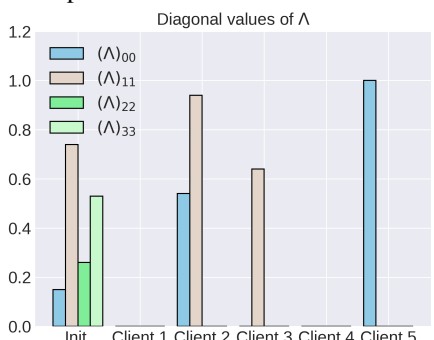

Figure 3: Diagonal values of $\{\Lambda_i\}$. All clients share the same initialization of $\{\Lambda_i\}$ (on the left). The right shows the learned $\{\Lambda_i\}$ in five clients by *Learn2pFed* under Setting 1.

**Impact of the Number of Layers on Convergence.** Figure 5 in Appendix C.1 shows the convergence of *Learn2pFed* in synthetic data w.r.t. three personalized settings mentioned above. It demonstrates that the deeper network, which unrolls more iterations $L$ of the ADMM, leads to faster convergence and more accurate solutions. Unless specified, we set $L = 10$ for the subsequent experiments.

### 5.3 POWER CONSUMPTION FORECASTING

**Dataset and Federated Settings.** We use the dataset Electricity Consuming Load (Lai et al., 2018) (ECL[1]) for electical load forecasting, which includes power consumption records (Kwh) for over 300 clients from 2011 to 2014. After data pre-processing, there are 313 candidate clients, each with 105216 records. We perform experiments following two personalized FL settings: (a) Setting 1 of full client participation scenario: we select 5 clients that have the most distinct properties, which are distinguished by using t-SNE technique (Van der Maaten & Hinton, 2008). (b) Setting 2 of partial client participation scenario: we randomly sample 50 clients to participate at each FL round.. In both cases, we split the local data of each selected client into train and test subsets in a ratio of 9:1.

**Results and Analysis.** We conduct five independent trials and report the averaged RMSE results evaluated on the testing dataset in Table 2. From the table, we see that 1) different from the results in regression simulation task, optimization-based methods, including pFedMe (T Dinh et al., 2020) and lp_proj (Lin et al., 2022), fail to perform well in such real-world complicated datasets and require large tuning efforts. In contrast, *Learn2pFed* still outperforms the other approaches with lower RMSEs. Additionally, we provide visualizations of the prediction results for both participating and non-participating clients in Appendix D.2.2 to verify the performance of the proposed *Learn2pFed*.

### 5.4 IMAGE CLASSIFICATION

**Dataset and Federated Settings.** We use two classical image classification datasets in FL, CIFAR-10 (Krizhevsky et al., 2009) and Fashion-MNIST (FMNIST) (Xiao et al., 2017) in two personalized settings: 1) we consider full client participation with $M = 10$ clients using the Dirichlet distribution (Yurochkin et al., 2019) with argument $\beta_{dir} = \{0.1, 0.5\}$, where a smaller $\beta_{dir}$ indicates the greater heterogeneity among the clients. 2) we consider partial client participation in order to follow the convention in federated learning literature, e.g., in Ditto and FedRep, where we perform the experiments with 100 clients, and 10 clients of them are chosen randomly per round. We further split the local data into training and testing sets at the ratio of 8:2 in both settings. In order to leverage the powerful representation capabilities of the deep neural network, we use the features extracted from the second-to-last layer of a CNN as the input of *Learn2pFed*. As a result, *Learn2pFed* aims to

---

[1]https://archive.ics.uci.edu/ml/datasets/ElectricityLoadDiagrams20112014

Table 2: Averaged RMSE for power consumption fore-casting task. Lower is better. Our proposed *Learn2pFed* consistently performs the best.

| Methods | Setting 1 | Setting 2 |
|---|---|---|
| Local-Only | $0.0998 \pm 0.0002$ | $0.2166 \pm 0.0001$ |
| FedAvg | $0.3796 \pm 0.1535$ | $0.7465 \pm 0.2454$ |
| FedProx | $0.3799 \pm 0.1533$ | $0.7471 \pm 0.2448$ |
| FedPer | $0.0342 \pm 0.0177$ | $0.1181 \pm 0.0737$ |
| FedRep | $0.0341 \pm 0.0178$ | $0.1182 \pm 0.0737$ |
| pFedMe | $0.0522 \pm 0.0276$ | $0.1226 \pm 0.0413$ |
| Ditto | $0.0339 \pm 0.0151$ | $0.1168 \pm 0.0754$ |
| lp_proj | $0.0733 \pm 0.0550$ | $0.1577 \pm 0.1200$ |
| *Learn2pFed* | $\mathbf{0.0307 \pm 0.0001}$ | $\mathbf{0.0619 \pm 0.0001}$ |

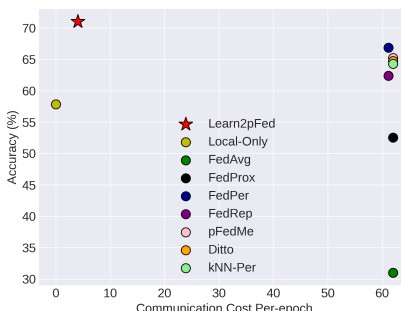

Figure 4: Communication cost and accuracy comparisons. *Learn2pFed* achieves the highest accuracy with minor communication cost.

linearly estimate the last fully-connected layer. Then, we jointly train the CNN and *Learn2pFed* with only the latter involved in FL communication.

**Results and Analysis.** 1) Table 3 shows that our proposed *Learn2pFed* consistently outperforms baselines across different datasets and different levels of data heterogeneity, indicating the effectiveness of learning to determine which parts of parameters for federation. 2) Figure 4 shows the communication cost per-epoch (KB) and accuracy of several representative methods in CIFAR-10 with 10 clients and $\beta_{dir} = 0.5$. Combined with results in Table 3, we see that our proposed *Learn2pFed* achieves the highest performance with minor communication cost, striking a great trade-off between communication cost and accuracy. This reveals another valuable property of our proposed *Learn2pFed* that it not only achieves pleasant accuracy, but also helps relieve communication cost. This reduction is attributed to *Learn2pFed* specifically replacing only the linear layers of the CNN model during federation, effectively minimizing communication overhead.

Table 3: Averaged classification accuracy (%) w.r.t. different Dirichlet parameters ($\beta_{dir}$) in CIFAR-10 and FMNIST in both settings, and communication (Comm.) cost (KB) per epoch. The proposed *Learn2pFed* consistently outperforms the state-of-the-art methods in both accuracy and Comm. cost, and the best results are in bold.

| Settings | 10 clients | | | | 100 clients | | | | Comm. cost (KB) | |
|---|---|---|---|---|---|---|---|---|---|---|
| Dataset | CIFAR-10 | | FMNIST | | CIFAR-10 | | FMNIST | | CIFAR-10 | FMNIST |
| $\beta_{dir}$ | 0.1 | 0.5 | 0.1 | 0.5 | 0.1 | 0.5 | 0.1 | 0.5 | - | - |
| Local-Only | 85.60 | 57.82 | 92.26 | 87.95 | 71.25 | 50.43 | 92.20 | 87.46 | 0 | 0 |
| FedAvg | 30.05 | 31.01 | 76.04 | 77.87 | 30.69 | 40.37 | 84.86 | 83.24 | 62.01 | 10.29 |
| FedProx | 41.68 | 52.54 | 80.42 | 86.19 | 52.56 | 48.22 | 90.84 | 87.13 | 62.01 | 10.29 |
| FedPer | 89.12 | 66.84 | 96.55 | 91.67 | 84.08 | 64.10 | 97.54 | 90.88 | 61.16 | 5.28 |
| FedRep | 86.56 | 62.39 | 96.03 | 88.72 | 84.81 | 60.27 | 96.60 | 90.11 | 61.16 | 5.28 |
| pFedMe | 90.31 | 65.19 | 97.48 | 92.86 | 83.11 | 51.07 | 98.15 | 88.56 | 62.01 | 10.29 |
| Ditto | 87.30 | 64.72 | 96.57 | 90.34 | 83.60 | 54.87 | 97.23 | 89.24 | 62.01 | 10.29 |
| CFL | 87.35 | 64.29 | 96.89 | 90.31 | 88.15 | 51.90 | 95.48 | 89.60 | 62.01 | 10.29 |
| kNN-Per | 88.47 | 64.28 | 97.64 | 90.09 | 74.69 | 61.74 | 92.13 | 88.82 | 62.01 | 10.29 |
| *Learn2pFed* | **90.71** | **71.02** | **98.06** | **94.09** | **89.45** | **71.64** | **98.97** | **91.99** | **4.06** | **4.06** |

# 6 CONCLUSION

We introduce *Learn2pFed*, a novel framework for personalized federated learning through algorithm unrolling. Our framework tackles the challenge of learning hyper-parameters that are typically unlearnable in the optimization process. By allowing the learnable parameters to determine the participation of local models in federated learning, we enhance adaptability of personalized FL methods. Extensive experiments on synthetic, time-series, and natural image datasets demonstrate the superior performance of *Learn2pFed*. Furthermore, as the unrolling-based framework, it holds potential for application in various scenarios in personalized FL approaches.

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
