## A    MORE DETAILS OF THE PROPOSED ALGORITHM

This section introduces the detailed updates of learnable parameters in Section 4.4 and the overall algorithm of the proposed *learn2pFed*.

### A.1    UPDATES OF LEARNABLE PARAMETERS

Recall from that $\mathcal{P}_b(\{\Lambda_i, p_i, \rho_i\})$ in (13) is the sum of local training losses collected by the server, the learnable parameters $\{\Lambda_i, p_i, \rho_i\}$ are updated using the gradient descent in the final $L$-th cell of the network as follows, mathematically.

$$
\begin{aligned}
p_i^L &= p_i^{L-1} - lr * \frac{\partial \mathcal{P}_b(\{\Lambda_i^{L-1}, p_i, \rho_i^{L-1}\})}{\partial p_i}, \\
\rho_i^L &= \rho_i^{L-1} - lr * \frac{\partial \mathcal{P}_b(\{\Lambda_i^{L-1}, p_i^L, \rho_i\})}{\partial \rho_i}, \\
\Lambda_i^L &= \Lambda_i^{L-1} - lr * \frac{\partial \mathcal{P}_b(\{\Lambda_i, p_i^L, \rho_i^L\})}{\partial \Lambda_i},
\end{aligned}
\tag{15}
$$

where the learning rate $lr$ can be set empirically. Note that we denote $\gamma_i^L$ as the copy of $\rho_i^L$ in the server, then the update of $\gamma_i^L$ is same as that of $\rho_i^L$ in (15). That is, $\rho_i^L$ is updated twice in both server side and client sides. To sum up, the global learnable parameters $\{p_i, \gamma_i\}$ are first updated in the server side as (15), and then the server broadcast the sum of losses $\mathcal{P}_b(\{\Lambda_i^{L-1}, p_i^L, \rho_i^L\})$ to each local client. The local learnable parameters $\{\Lambda_i, \rho_i\}$ are then updated in the client sides as (15).

### A.2    ALGORITHM

The overall algorithm is displayed as below.

---

**Algorithm 1** *Learn2pFed*: layer-wise training.

---

**Input:**  The number of local clients $M$; the number of ADMM iterations $L$; the maximum epoch $E$ for training.
**Output:**  Personalized local model $\{v_i^L\}_{i \in [M]}$.
 1:  Initialize personalized models $\{v_i^0, z_i^0, \alpha_i^0\}$ randomly, and global model $w^0$. Initialize learnable parameters $\{\rho_i^0, \gamma_i^0, \Lambda_i^0, p_i^0\}$.                              ▷ Initialization
 2:  **for** $e = 1$ to $E$ **do**
 3:      **for** $\ell = 1$ to $L$ **do**
 4:          **for** $i = 1$ to $M$ (parallel) **do**                              ▷ Client-Side Computation
 5:              Update $\alpha_i^\ell$ via (9).
 6:              Update $v_i^\ell$ via (10).
 7:              Update $z_i^\ell$ via (11).
 8:              Send vector $vec = v_i^\ell - z_i^\ell - \alpha_i^\ell$ to the server.
 9:          **end for**
10:          Update $w^\ell$ via (12) and broadcast it to the local.              ▷ Server-Side Computation
11:      **end for**
12:      The server collects each local training loss $\mathcal{L}_i(X_i v_i^L, Y_i)$ in (13) and updates the global learnable parameters $\{p_i^L, \gamma_i^L\}$ via (15). Then, the server broadcasts the sum of losses back to the clients.                              ▷ Global Learnable Parameters Update
13:      Each local client receives the losses from the server, and update the learnable parameters both in the client sides via (15).                              ▷ Local Learnable Parameters Update
14: **end for**
15: **return** $\{v_i^L\}$ after $E$ epochs.

---

### A.3    MORE DISCUSSIONS

In spite of the privacy, we have discussed the memory and computation cost in this subsection.

**Practical resource usage.** We present the memory and computation cost of *Learn2pFed* in CIFAR10 classification, comparing them to those of FedAvg. For FedAvg model, its feature extractor occupies 11.19 KB, and the three linear layers occupy 187.03 KB, 39.84 KB, and 3.32 KB, respectively. Hence, the total memory consumption is mainly in the storage of the linear layers. And its FLOPs are 0.6517M. In our collaborative training, the feature extractor occupies the same 11.19 KB, and the *Learn2pFed* network only occupies 12.69 KB and cost 0.6517M FLOPs. Thus, due to the replacement of the last few linear layers of the network with *learn2pFed* in the classification task, our method saves 94.49% memory cost excluding the feature extractor and 90.11% including the feature extractor, and a 9.04% reduction in FLOPs. However, the magnitude of these advantages will gradually decrease as the local feature extractor grows larger. This is because *learn2pFed* is applied primarily to the last few layers, and the reduction in memory brought about by these layers becomes relatively smaller compared to the memory occupied by the feature extractor.

**Memory/computation complexity.** Unrolling introduces a modest increase in memory and computation complexity compared to standard approaches. Our experiments show that this additional overhead is manageable. However, for larger model architectures, the effect on resource usage may become more noticeable.

## B DETAILED EXPERIMENTAL SETUP

**Hardware and software.** All the experiments are implemented in PyTorch and simulated in NVIDIA GeForce RTX 3090 GPUs. Core codes are available in the anonymous link[2].

**Models.** In the simulation for regression, a simple 5-layer MLP model with output size of each layer as $[200, 100, 50, 10, 1]$. In the real-data experiments, we consider a long short-term memory (LSTM (Yu et al., 2019)) model with two hidden layers, followed by a fully-connected layer for power consumption forecasting. It processes a sequence of length 3, and its hidden states have Tanh activations with 12 dimensions. And the loss function is Mean-Squared-Error (MSE) loss. Besides, we utilize two different CNN models for classification in CIFAR-10 and Fashion-MNIST, respectively. Both of them are constructed by two convolution layers, each followed by a max pooling layer, and three and one fully-connected (fc) layer, respectively, before softmax output. The convolution channel and the output unit of fc are $[6, 16]$ and $[128, 84, 10]$ for CIFAR-10, $[10, 20]$ and $[10]$ for Fashion-MNIST. The LeakyReLU (Xu et al., 2020) is used as the activation function, and the loss function is Cross-Entropy (CE) loss.

## C MORE NUMERICAL RESULTS

This section shows the numerical results of the ablation studies in Section 5.2 and 5.4, whose main conclusions has been represented in the paper.

### C.1 MORE NUMERICAL RESULTS IN SYNTHETIC DATA

**Impact of learnable parameters in synthetic data.** We aim to investigate which specific learnable parameters play a more important role in the proposed *Learn2pFed* by repeating the simulations for five times and show the results in Table 4. It shows that learning more parameters perform better. Specifically, learning $\{\Lambda_i\}$ indicating the adaptive federated collaboration is significantly effective.

---

[2]https://anonymous.4open.science/r/Learn2pFed-DD63

Table 4: Learnable parameters ablation study w.r.t. three personalized FL settings on synthetic data: we leave the check marks under the learnable parameters and the blanks under the non-learnable parameters. The experiments with more learnable parameters perform better.

| Learnable parameters | | | | | | Averaged RMSE of *Learn2pFed* | | |
|---|---|---|---|---|---|---|---|---|
| $\{\Lambda_i\}$ | $\{p_i\}$ | $\{\rho_i\}$ | $\{\eta_i\}$ | $\{\gamma_i\}$ | $\{\theta_i\}$ | Setting 1 | Setting 2 | Setting 3 |
| ✓ | | | | | | $0.0007 \pm 0.0001$ | $0.0010 \pm 0.0001$ | $0.0008 \pm 0.0002$ |
| | ✓ | | | | | $0.0018 \pm 0.0001$ | $0.0226 \pm 0.0001$ | $0.0100 \pm 0.0003$ |
| ✓ | ✓ | | | | | $0.0013 \pm 0.0001$ | $0.0008 \pm 0.0002$ | $0.0058 \pm 0.0005$ |
| | | ✓ | ✓ | ✓ | ✓ | $0.0026 \pm 0.0024$ | $0.0007 \pm 0.0003$ | $0.0015 \pm 0.0003$ |
| | | ✓ | | | | $0.0084 \pm 0.0006$ | $0.0544 \pm 0.0037$ | $0.0302 \pm 0.0154$ |
| | | | ✓ | | | $0.0037 \pm 0.0006$ | $0.0016 \pm 0.0004$ | $0.0051 \pm 0.0000$ |
| | | | | ✓ | | $0.0126 \pm 0.0009$ | $0.0059 \pm 0.0013$ | $0.0524 \pm 0.0005$ |
| | | | | | ✓ | $0.0040 \pm 0.0004$ | $0.0023 \pm 0.0002$ | $0.0045 \pm 0.0021$ |
| | ✓ | ✓ | ✓ | ✓ | ✓ | $0.0085 \pm 0.0071$ | $0.0084 \pm 0.0038$ | $0.0032 \pm 0.0019$ |
| ✓ | | ✓ | ✓ | ✓ | ✓ | $0.0018 \pm 0.0001$ | $0.0046 \pm 0.0034$ | $0.0009 \pm 0.0003$ |
| ✓ | ✓ | ✓ | ✓ | ✓ | ✓ | $0.0002 \pm 0.0002$ | $0.0003 \pm 0.0002$ | $0.0003 \pm 0.0002$ |

**Impact of the number of layers on convergence in synthetic data.** We investigate the impact of varying values of the ADMM iterations $L$ on the convergence. Additionally, we illustrate the loss over the previous one hundred epochs as part of our analysis in Figure 5. It indicates that as $L$ increases, the convergence becomes more stable and the obtained solution becomes more accurate. However, it also results in more communication rounds of *Learn2pFed*. In this trade-off between performance and communication overhead, we can choose $L = 10$ as a suitable balance.

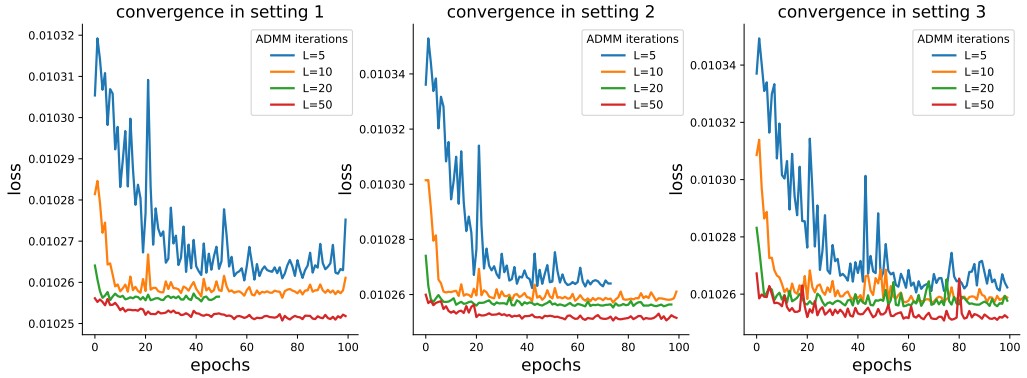

Figure 5: ADMM iteration ablation study. It shows that the deeper *Learn2pFed* has better convergence.

### C.2 MORE NUMERICAL RESULTS IN IMAGE CLASSIFICATION DATA

**Impact of layers where *Learn2pFed* starts in CIFAR-10.** We conduct an ablation study on the selection of features extracted from the layer before the specific fully-connected layers as inputs to *Learn2pFed*, i.e., where *Learn2pFed* starts. This experiment is performed in the CIFAR-10 classification task, where $M = 10$, and $\beta_{dir} = 0.1$. Table 5 shows the quantitative results. In this study, we introduced the concept of a shared ratio, which represents the ratio of shared parameters to local parameters. When *Learn2pFed* starts from the first fc layer, the shared ratio increases significantly because the basic CNN's parameter quantity primarily concentrates on the first fc layer. For this reason, our method may not always demonstrate a significant ability to reduce communication costs when starting at different layers. However, it does not impact much on the accuracy (see in Figure 6).

Table 5: Ablation study on layers where *Learn2pFed* starts. The shared ratio denotes the ratio of local parameters to shared parameters where the latter is fixed in *Learn2pFed*. The fully-connected layers' input and output dimension ($dim$) follows the architecture introduced in Appendix B.

| fully-connected layers | | | local parameters (KB) | shared ratio (%) |
|---|---|---|---|---|
| first $dim$ [400,120] | second $dim$ [120,84] | third $dim$ [84,10] | | |
| | | ✓ | 61.14 | 14.72 |
| | ✓ | | 50.99 | 17.65 |
| ✓ | | | 2.87 | 313.58 |

# D    MORE VISUAL ILLUSTRATIONS

In this section, we present the visual illustrations to show the performance in regression (Section 5.2) and forecasting (Section 5.3) tasks. The local data distribution illustration in classification (Section 5.4) is also provided in this section.

## D.1    MORE VISUAL ILLUSTRATIONS IN REGRESSION IN SEC. 5.2

**Detailed implementation.** We take the 3-rd polynomial federated regression in setting 1 on synthetic data for example, where the polynomial coefficient vector is $\boldsymbol{a} = [0, -6, 18, -12]$. And we sample 1000 points from the ground-truth and add the Gaussian noise according to the context in Sec. 5.2. The data points are randomly scattered to $M = 5$ clients in the ascending order of the values in x-axis, with local clients has $[58, 172, 278, 233, 256]$ samples, respectively.

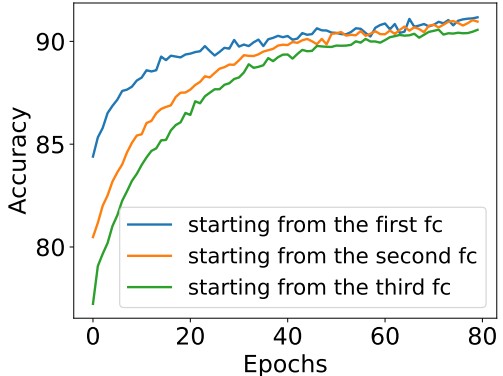

Figure 6: Accuracy of *Learn2pFed* in CIFAR-10 with $\beta_{dir} = 0.1$. Where *Learn2pFed* starts has little impacts on the accuracy in classification in CIFAR-10.

**Visualization.** We take the simulations in setting 1 in synthetic data for example, and show the visualizations of the regression performance in Figure 7. The results indicate that a single global model from generalized FL methods such as FedAvg and FedProx cannot achieve good performance. In addition, their finetuning-based version and independent learning (IL) method can be easily disturbed by the local data with Gaussian noise. On the other hand, optimization-based methods perform well in this scenario. The proposed *Learn2pFed* stands out as the best fit for a three-order polynomial model.

## D.2    MORE VISUAL ILLUSTRATIONS IN FORECASTING IN SEC. 5.3

In this sub-section, we mainly show the visual illustrations in the power consumption forecasting task in Section 5.3 with ECL dataset in setting (a) where five clients are chosen. Specifically, we will show the performance on one of participating clients and one non-participating client.

### D.2.1    DATA PRE-PROCESSING AND CHOOSING THE PARTICIPATING LOCAL CLIENTS
We first perform the data pre-processing for ELD, including giving up the clients with extremely high consumption and the missing data. Then, we select a time period after January 1st 2012. Following the instruction[3], we first convert the ELD data values in kWh values by dividing them by 4. The local clients with outliers and missing values are then excluded based on average electricity usage. Second, we use the Elbow Method (Bholowalia & Kumar, 2014) to determine the number of clusters and perform the t-SNE (Van der Maaten & Hinton, 2008) with four clusters, which visualization is shown in Figure 8. Finally, we choose the five targets named as MT_001, MT_002, MT_003,

---

[3]https://archive.ics.uci.edu/ml/datasets/ElectricityLoadDiagrams20112014

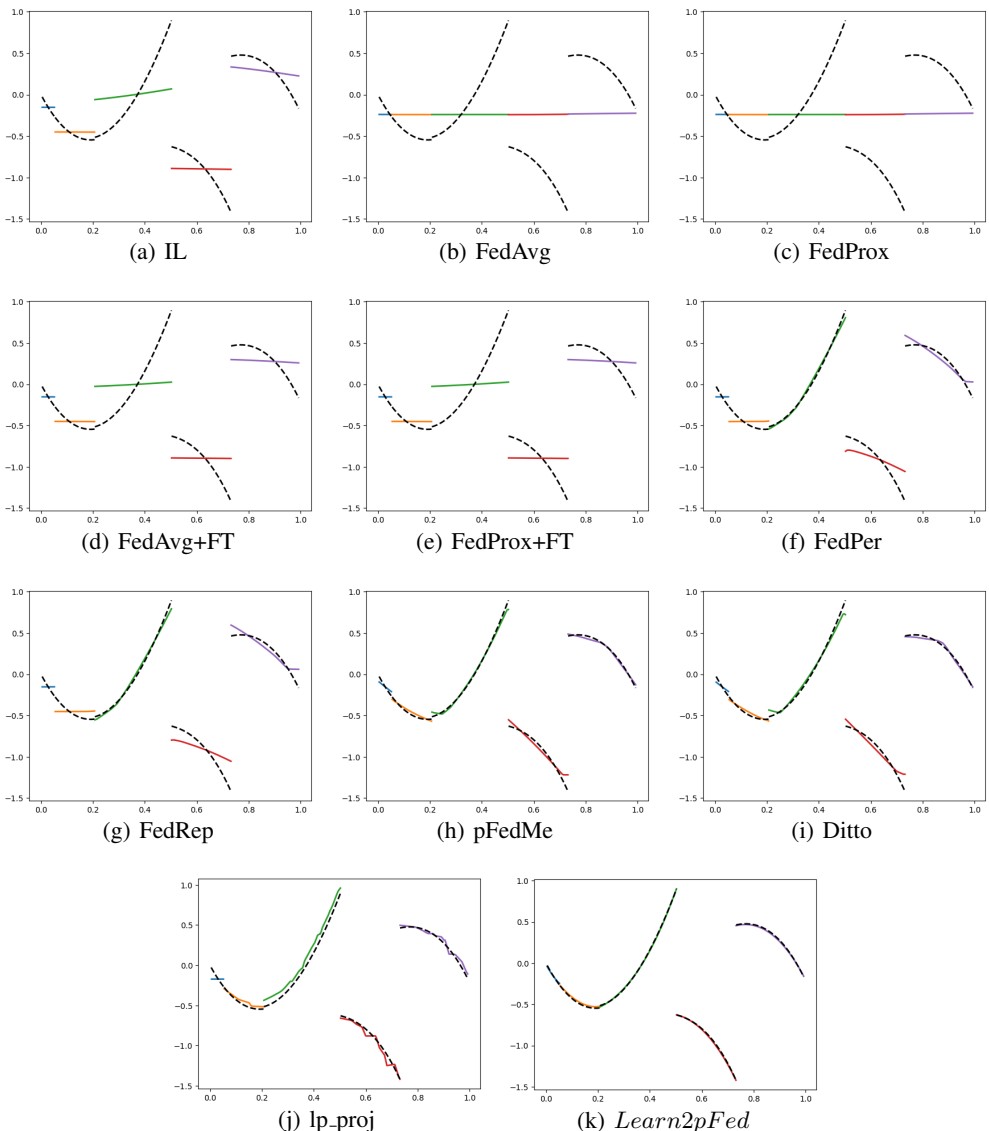

Figure 7: Visualization of the regression performance w.r.t. different (personalized) federated learning methods under setting 1: the black dot lines denote the local ground-truth and the colored lines depict the fitted models. Our *Learn2pFed* fits the three-order polynomial model best.

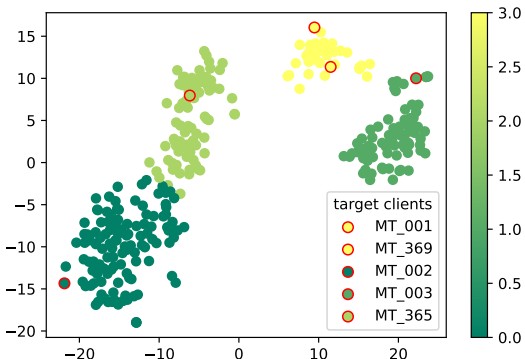

Figure 8: The visualization of ELD features by t-SNE, and the five local clients used in our experiment.

MT_365 and MT_369 for personalized FL training. In this way, we guarantee the personalization in the participating local clients.

### D.2.2 MORE VISUAL ILLUSTRATIONS IN FORECASTING WITH ECL IN SETTING (A)

We compare the proposed *Learn2pFed* with the baseline methods, and depict the visualization of the participating client named as MT_002 in setting (a) in Figure 9, and that of the non-participating client named as MT_014 in Figure 10. Note that we only illustrate the prediction results of the first two thousand data points of the testset. From Figure 9 and 10, *Learn2pFed* achieves best forecasting performance.

### D.3 MORE VISUAL ILLUSTRATIONS IN CLASSIFICATION IN SEC. 5.4

**Visualization of local data distribution in classification tasks in personalized FL.** To generate the personalized data, we use the Dirichlet distribution (Yurochkin et al., 2019) with hyper-parameters $\beta_{dir} = \{0.1, 0.5, 5.0\}$, which is a widely considered setting (Marfoq et al., 2022; Hsu et al., 2019). We depict the local data distribution of the training samples in CIFAR-10 in Figure 11 across $M = 10$ clients. It is shown that as $\beta_{dir}$ increases, the local data coverage encompasses a more comprehensive range of sample categories, and the number of samples in each category becomes more balanced.

## E LIMITATION AND FUTURE WORK

*Learn2pFed* focuses on dynamically determining the local parameters that should participate in the federated collaboration, but a limitation arises in its ability to explain the physical meaning of those parameters selected by*Learn2pFed*. We are curious whether it can provide some insight for model compression or data selection. Therefore, we aim to further explore this in future work.

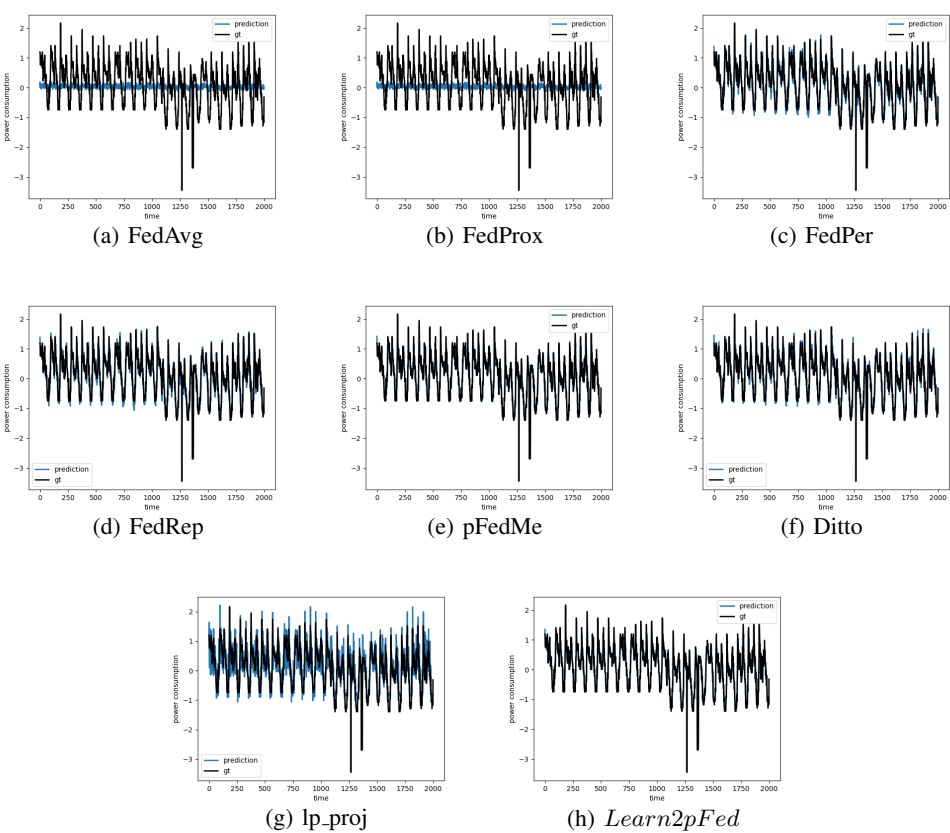

Figure 9: Visualization of the forecasting performance w.r.t. different (personalized) federated learning methods on the participating client MT_002 in ECL data under setting (a). The black lines denote the groud-truth of the testset, and the blue lines denote the prediction results. Our *Learn2pFed* achieves best forecasting performance.

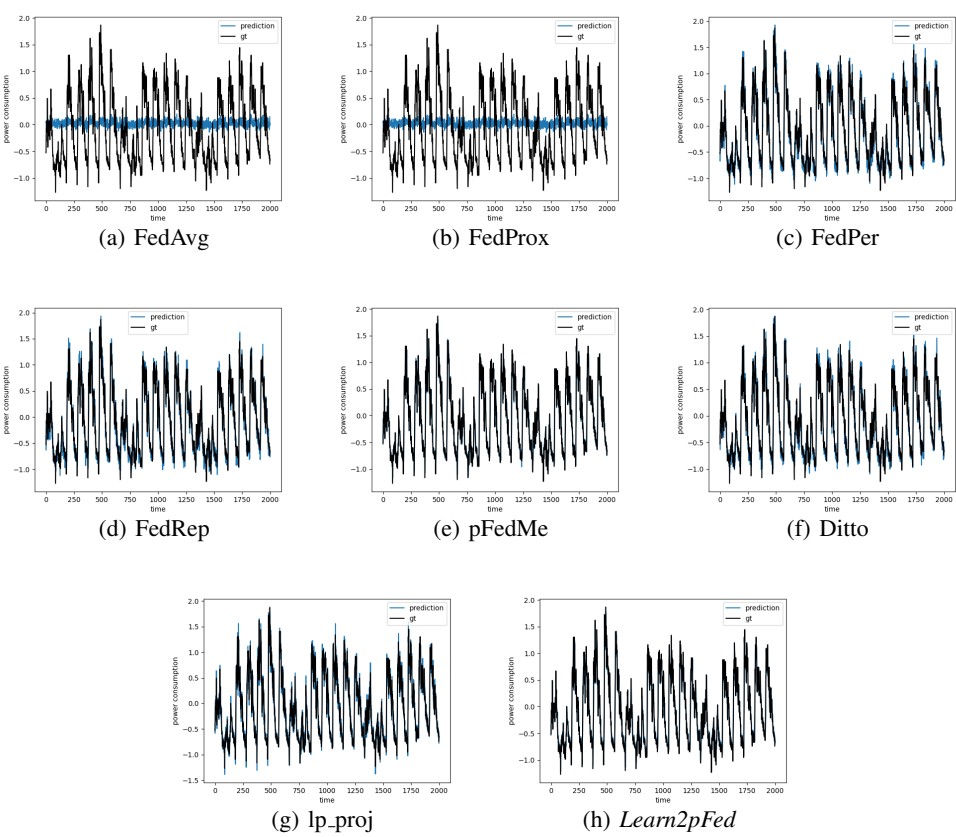

Figure 10: Visualization of the forecasting performance w.r.t. different (personalized) federated learning methods on the non-participating client MT_014 in ECL data under setting (a). The black lines denote the groud-truth of the testset, and the blue lines denote the prediction results. Our *Learn2pFed* achieves best forecasting performance.

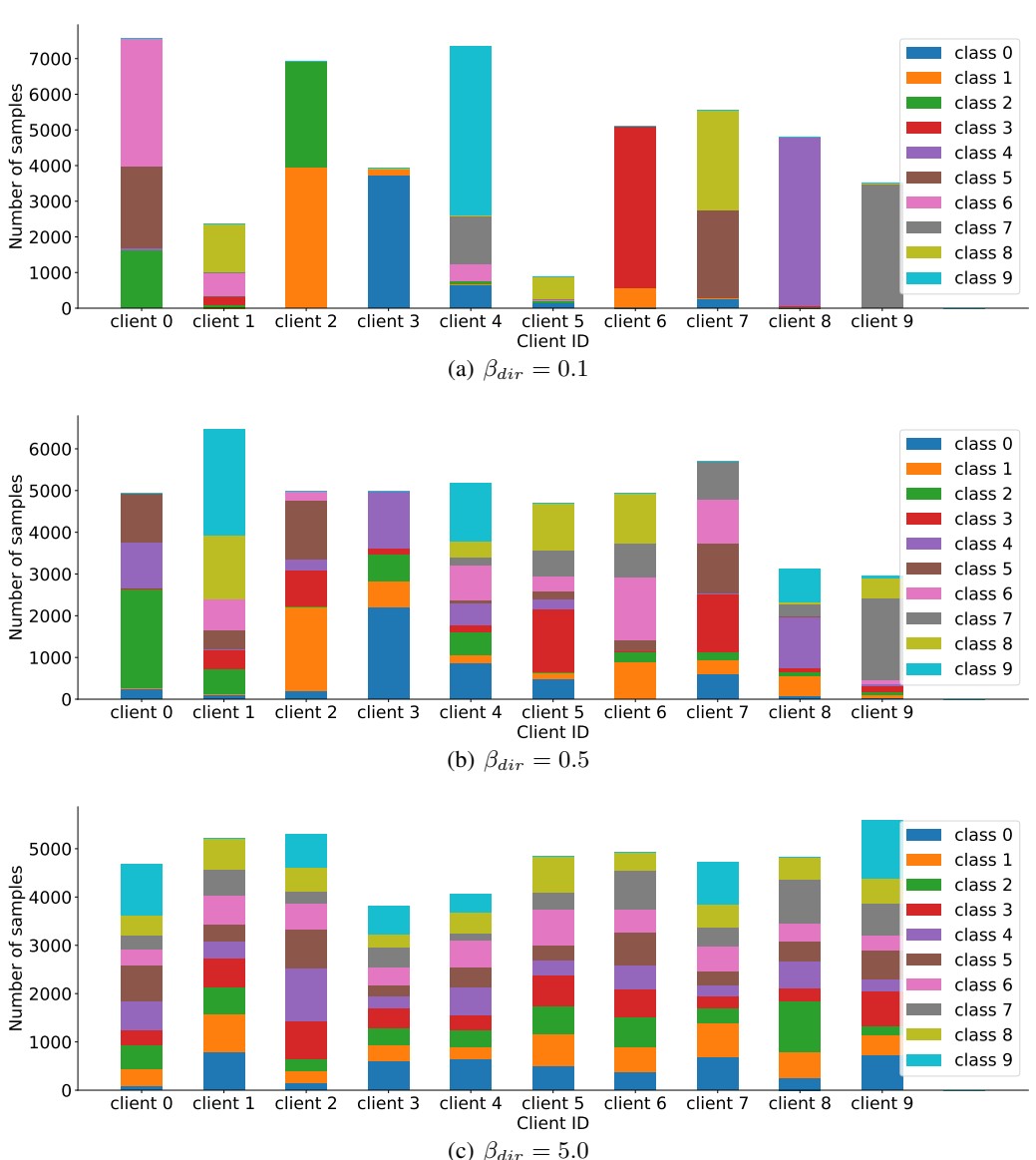

Figure 11: Visualization of local data distribution in classification tasks in CIFAR-10 w.r.t. different Dirichlet parameter $\beta_{dir}$. A smaller $\beta_{dir}$ indicates the greater heterogeneity among the clients.