# OpenReview forum: "Learn What You Need  in Personalized Federated Learning"
_ICLR.cc/2024/Conference — ICLR 2024 Conference Withdrawn Submission_

### Official Review · Reviewer_pvjk · 2023-10-30

**Soundness:** 2 fair
**Presentation:** 3 good
**Contribution:** 2 fair
**Rating:** 3
**Confidence:** 4

**Summary:**

In this paper, the authors aim to enhance the adaptive aggregation in a customized way for personalized federated learning. They proposed an algorithm-unrolling-based method, i.e., Learn2pFed, to adaptively choose the part of parameters and the degree in aggregation. To validate Learn2pFed, they conduct extensive tasks, i.e., regression, forecasting, and image classification.

**Strengths:**

Strengths:
I enjoy the insights of this work for (1) clear presentation, (2) methodology design, and (3) the extensive experiments.

S1: The authors present this paper with clear illustration and systematic logic outline for optimizing global model and client models.
S2: The mechanism of enhancing both the parts and the degree of model parameters is reasonable for enhancing performance. Besides, the proposed Learn2pFed method learns hyper-parameters via algorithm unrolling, which is more flexible.
S3: It interests us for conducting extensive experiments in different tasks, i.e., regression, forecasting, and image classification.

**Weaknesses:**

I feel it uncertain and weak from the aspects of: (1) the theoretic analysis, (2) the privacy leakage, (3) the novelty of customized aggregation, and (4) the reproducibility.
W1: The convergence bound of Learn2pFed is not provided, and complexity analysis related to stability is deficient, which is necessary to clarify the concerns of additional computation and memory burden, since in Fig.5 appendix relies on more than 50 iterations.
W2: The privacy enhancement is limited. Iterative optimization, e.g., ADMM, relies on the gradient exchange among clients and servers. And the authors claim that a linear combination of multiple local variables can achieve the goal of privacy-preserving. However, no empirical and theoretic analysis is present.
W3: The related work is insufficiently studied in empirical results. Personalized and adaptive aggregation have been studied in (1) choosing model parameters parts[3,4], and (2) adaptive the degree of model aggregation [1,2,5]. It is necessary to compare the difference among these methods for highlighting the contribution of considering both.

[1] Li Z, Lin T, Shang X, et al. Revisiting weighted aggregation in federated learning with neural networks[J]. arXiv preprint arXiv:2302.10911, 2023.
[2] Zhang J, Hua Y, Wang H, et al. FedALA: Adaptive local aggregation for personalized federated learning[C]//Proceedings of the AAAI Conference on Artificial Intelligence. 2023, 37(9): 11237-11244.
[3] Lu W, Hu X, Wang J, et al. FedCLIP: Fast Generalization and Personalization for CLIP in Federated Learning[J]. arXiv preprint arXiv:2302.13485, 2023.
[4] Isik B, Pase F, Gunduz D, et al. Sparse random networks for communication-efficient federated learning[J]. arXiv preprint arXiv:2209.15328, 2022.
[5] Liao, Xinting, et al. "HyperFed: Hyperbolic Prototypes Exploration with Consistent Aggregation for Non-IID Data in Federated Learning." arXiv preprint arXiv:2307.14384 (2023).

W4: The implementation code is not open-source, which brings me three concerns, i.e., (1) the performance generalization of Learn2pFed in more complex datasets, (2) the additional computational burden for the newly proposed method, and (3) the capability of privacy in attack.

**Questions:**

Q1: Since the existing analysis of convergence, computation and memory are all depended on the empirical studies with regard to small datasets, could the authors provide us with more theoretic analysis.
Q2: Could the authors provide us with related privacy defense analysis?
Q3: How can Learn2pFed become more scalable to large and complex federated setting, which is more practical in real-world applications?

---

### Official Review · Reviewer_tbfK · 2023-10-30

**Soundness:** 3 good
**Presentation:** 3 good
**Contribution:** 3 good
**Rating:** 6
**Confidence:** 3

**Summary:**

The paper proposes a new personalized federated learning algorithm named Learn2pFed. Instead of aggregating the full model parameters in each round, the paper proposes to adaptively set the degree of participation of each model parameter by learning additional variables. These variables are optimized by leveraging algorithm unrolling. Experiments on regression and classification tasks demonstrate that Learn2pFed outperforms the other baselines.

**Strengths:**

1. The idea of learning the degree of participation of each model parameter is promising.

2. Leveraging algorithm unrolling to optimize the hyperparameters is interesting.

3. Experiments are comprehensive. Three different tasks are covered and the improvement of Learn2pFed is significant.

**Weaknesses:**

1. One concern is that Learn2pFed needs to update and transfer the parameters at a layer level. Compared with the model aggregation method, the communication frequency of Learn2pFed is much higher especially when the model is deep.

2. The paper does not provide a theoretical convergence analysis of Learn2pFed.

3. The learning process is a bit complicated as six additional learnable parameters are introduced.

**Questions:**

1. In Training Detailes, the paper claims that 500 communication rounds are adopted. For Learn2pFed, I think a communication round refers to the whole Algorihtm 1, where in fact many communication rounds happen (i.e., $E*L$). Am I right?

2. What is the elapsed training time of Learn2pFed? I'm curious whether learning the introduced parameters will incur much computation overhead.

3. In Table 2 and Table 3, FedAvg + FT and FedProx + FT are not presented. Why do not keep the baselines consistent in the experiments?

---

### Official Review · Reviewer_BaaV · 2023-10-31

**Soundness:** 2 fair
**Presentation:** 3 good
**Contribution:** 2 fair
**Rating:** 5
**Confidence:** 4

**Summary:**

This paper falls into the personalized federated learning domain. To personalize the model training, the authors design an approach to partially update the local model parameters. It provides experiment results on different types of tasks and compares them with selected baselines.

**Strengths:**

1. The paper is easy to follow and read.
2. The authors discuss the power consumption of the proposed algorithm and make comparisons with existing work.

**Weaknesses:**

1. I am concerned about the motivation of this work. Personalized federated learning is an interesting but not new research track, where there are already many existing works in this domain. Though discussed in the related works, I am still confused about the advantages of this algorithm and the motivation for combining unrolling with federated learning.
2. There are several partial model update studies in FL, for example: [1-3]. I am curious about the main differences and advantages compared with the works that I mentioned and the related works covered in this paper.
3. Though it provides discussions about privacy leakage, I am concerned about the extra information exchanged between the server and the clients.

[1] Singhal, Karan, Hakim Sidahmed, Zachary Garrett, Shanshan Wu, John Rush, and Sushant Prakash. "Federated reconstruction: Partially local federated learning." Advances in Neural Information Processing Systems 34 (2021): 11220-11232.

[2] Sun, Guangyu, Matias Mendieta, Jun Luo, Shandong Wu, and Chen Chen. "FedPerfix: Towards Partial Model Personalization of Vision Transformers in Federated Learning." In Proceedings of the IEEE/CVF International Conference on Computer Vision, pp. 4988-4998. 2023.

[3] Sun, Benyuan, Hongxing Huo, Yi Yang, and Bo Bai. "Partialfed: Cross-domain personalized federated learning via partial initialization." Advances in Neural Information Processing Systems 34 (2021): 23309-23320.

**Questions:**

1. If the clients’ model structure is large and more complicated, would the computation burden on the client side be a problem?
2. Please clarify the motivation and benefits of using this technique in the personalized FL compared with other PFL approaches.
3. What if we have 100 or 200 clients, which is a typical setting in FL? Do we have a different participant ratio at each communication round? Please specify the scalability of this algorithm.

---

### Official Review · Reviewer_7Lgr · 2023-10-31

**Soundness:** 3 good
**Presentation:** 3 good
**Contribution:** 3 good
**Rating:** 6
**Confidence:** 5

**Summary:**

A personalized federated learning method called Learn2pFed is proposed. Learnable parameters are used to control the personalization of each weight in the model, so that the personalized part and the degree of personalization in PFL can be controlled more accurately. The effectiveness of this method is verified by comparing it with several PFL methods on regression, prediction and classification tasks.

**Strengths:**

1.Learnable hyperparameters are adopted to adaptively control the scope and degree of personalization.

2.The proposed method controls the degree of personalization of each parameter, which has a finer granularity than other methods, and can control personalization more accurately.

**Weaknesses:**

Methodology:

1.Since more trainable parameters have been added in the method, there are additional communication overheads in computation and communication.

2.The proposed method essentially increases the number of trainable parameters, i.e., the model capacity. This method in fact affects the fairness of the comparison of other methods, and the performance of different methods should be compared with the same number of learnable parameters.

Writing:

1.Introduction 2nd paragraph 1st line "personalized Federated Learning (FL)" abbreviation appears redundantly (1st line of 1st paragraph)

2.In section 4.4, paragraph 3, fourth line: "serveri n"->"server in"

**Questions:**

Would it be possible to add a comparison of the communication and computational overhead of the different methods?